# Effects of Short-Term Differences in Concentrate Feeding on the Recovery of In Vivo Embryos in Hanwoo Donor Cows through Superovulation Treatment

**DOI:** 10.3390/ani14172591

**Published:** 2024-09-06

**Authors:** Seungmin Ha, Namtae Kim, Mi-Ryung Park, Seyoung Lee, Sang-Rae Cho, Huimang Song, Daehyeok Jin, Ui-Hyung Kim, Yeoung-Gyu Ko

**Affiliations:** 1Animal Genetic Resources Research Center, National Institute of Animal Science, Rural Development Administration, Hamyang 50000, Republic of Korea; justusha@korea.kr (S.H.); namtea77@korea.kr (N.K.); mrpark45@korea.kr (M.-R.P.); sylee251@korea.kr (S.L.); chosr@korea.kr (S.-R.C.); hope0909@korea.kr (H.S.); jdh1662@korea.kr (D.J.); 2Hanwoo Research Institute, National Institute of Animal Science, Rural Development Administration, Pyeongchang 25340, Republic of Korea; uhkim@korea.kr

**Keywords:** superovulation, concentrate intake, Hanwoo, CIDR, short-term, embryos

## Abstract

**Simple Summary:**

Superovulation is a hormonal treatment used to produce multiple oocytes in cows simultaneously, which is essential for embryo transfer. Its success depends on various factors such as hormone levels, protocols, timing of insemination, weather, and nutrition. However, the impact of feed changes, specifically the energy levels in the concentrate, on embryo production is not well understood. This study explores how changes in concentrate intake, with unlimited access to hay, affect embryo production in indigenous Korean (Hanwoo) cows. We compared embryo production and physiological changes between cows with altered concentrate intake and those with consistent intake. The results offer new insights for managing superovulation in relation to feed concentrate.

**Abstract:**

Superovulation is a technique used to increase the number of oocytes released for fertilization. This study investigated the effects of short-term differences in concentrate feed intake on in vivo embryo production through superovulation in indigenous Korean (Hanwoo) cows. The cows were given fresh water and hay ad libitum and randomly divided into three groups (control (CON, n = 9): 2.0 kg/day (unchanged diet); low concentrate (LC, n = 10): 0 kg/day; and high concentrate (HC, n = 8): 4.0 kg/day) according to the amount of formula they were fed. This feeding treatment began seven days before the start of the hormonal treatment for superovulation. From the results, the LC group had the greatest weight change and the lowest body condition score at harvest, followed by the CON and HC groups (*p* < 0.05). The LC group had the highest number of harvesting embryos, followed by the HC and CON groups (*p* < 0.05). Estradiol, progesterone, glucose, total cholesterol, high-density lipoprotein, low-density lipoprotein, total protein, and blood urea nitrogen concentrations did not differ between the groups, except for a temporary increase in the HC group on day 0. These findings suggest that more embryos may be harvested when short-term changes in concentrate intake are made during superovulatory responses in Hanwoo cows.

## 1. Introduction

Superovulation and embryo transfer (ET) are widely used techniques in the cattle industry. The commercialization of these techniques has been gaining global popularity since the 1970s. These techniques were primarily developed to import European bovine breeds into North America [1]. According to Mikka et al., two methods, viz. in vivo-derived (IVD) embryos obtained through the use superovulation protocols, and in vitro-produced (IVP) embryos via ovum pick-ups, in vitro maturation, and in vitro fertilization are commonly used to reproduce genetically selected cattle [2]. Although the production of IVP embryos is higher than that of IVD embryos worldwide, IVD embryo production was represented by approximately 20% of the total production in 2018 [3]. Increasing the number of fertilized and transferable embryos with high potential for achieving pregnancy is the principal goal of superovulation in cattle [4].

Superovulation’s success is potentially influenced by several factors [2]. The ovarian response to superovulation can be predicted using anti-Müllerian hormone concentrations in cows. The success of superovulation is influenced by technical factors such as gonadotrophin level, superovulation protocol, timing of insemination, and semen quality. Extremely hot and cold temperatures and psychological stress-induced elevation of cortisol concentrations exacerbate superovulation.

Proper nutrition is crucial to achieve reproductive success. Energy deficiencies can lead to extended postpartum anestrus and lower conception rates due to impaired follicle development and ovulation [5]. Conversely, excess energy can cause obesity, leading to metabolic issues that also hinder reproduction [6]. Obesity in cattle is associated with increased insulin resistance and altered endocrine function, negatively impacting ovarian activity and fertility. However, the effects of nutritional intake in conjunction with superovulation in cattle remain unknown. According to previous studies, a negative energy balance in dairy cattle has been shown to have adverse effects on superovulation, whereas short-term nutritional restriction improves follicle growth and embryo production [2,7]. A high plane of energy for 7 weeks showed detrimental effects, whereas that for 3 weeks showed beneficial effects on superovulation in cattle [8,9]. In addition, unlimited concentrate intake negatively affected embryo yield and quality [10]. Previous studies have focused on the unlimited concentrate intake of other cow breeds, but no one has conducted a similar study on indigenous Korean (Hanwoo) cows.

We hypothesized that the limited amount of increasing or decreasing concentrate intake would significantly change embryo production during the superovulation. Therefore, we aimed to investigate the effects of short-term (23 days) concentrate-feeding changes on embryo production and physiological status in indigenous Korean (Hanwoo) cows compared to those fed an unchanged diet.

## 2. Materials and Methods

### 2.1. Animals and Experimental Design

A total of 27 cows raised on a housing farm of the National Institute of Animal Science in Republic of Korea were used in this study. They were fed 2.0 kg/day of formula feed (NongHyup feed, Haman-Gun, Republic of Korea) and were given fresh water and hay ad libitum. The cows were randomly categorized into three groups according to the amount of formula they were fed: control (CON, n = 9), 2.0 kg/day; low concentrate (LC, n = 10), 0 kg/day; and high concentrate (HC, n = 8), 4.0 kg/day (Table 1). No significant differences were observed among the CON, LC, and HC groups in terms of age, parity, days since calving, BCS, and body weight at the start of the study. Specifically, the cows in the CON group were 7.20 ± 0.82 years old, with 2.11 ± 0.33 parities, 333.7 ± 94.3 days since calving, a body condition score (BCS) of 3.28 ± 0.26, and an average body weight of 452.9 ± 51.9 kg. The cows in the LC group were 7.27 ± 1.27 years old, with 2.00 ± 0.47 parities, 273.8 ± 54.0 days since calving, a BCS of 3.15 ± 0.24, and an average body weight of 478.0 ± 46.5 kg. The HC group comprised cows that were 7.05 ± 1.13 years old, with 1.88 ± 0.35 parities, 283.8 ± 50.6 days since calving, a BCS of 3.38 ± 0.44, and an average body weight of 502.0 ± 85.9 kg. The formula feed consisted of ≤14% crude protein, ≤2.5% crude fat, ≥18% crude fiber, ≥10% crude ash, ≤0.7% calcium, ≥1.2% phosphorus, and 71% total digestible nutrients. The formula was given to the cows one week before the start of the hormonal treatment for superovulation (Figure 1). The formula feed was continued for 23 days until the embryos were harvested.

### 2.2. Estrous Synchronization and Artificial Insemination (AI)

Hormonal treatment for superovulation was performed on the experimental animals using a superovulation protocol modified for Hanwoo cows [11,12], as shown in Figure 1. Synchronization of estrous was confirmed in all cows (random stages of the estrous cycle) with the insertion of a progesterone-releasing intravaginal device for controlled internal drug release (CIDR; InterAg, Hamilton, New Zealand) in the vagina. The device was inserted 7 d after the beginning of the imposed dietary treatments. The cows were given 1 mg estradiol (Estron, Samyang Anifarm, Seoul, Republic of Korea) and 50 mg progesterone (Ovarone, Samyang Anifarm, Seoul, Republic of Korea) immediately after the CIDR device was inserted (9 am, Day 0). The cows were superovulated with follicle-stimulating hormone (FSH; Antorin R-10, Kawasaki Mitaka, Japan), which was administrated at 9 am and 9 pm on days 4, 5, 6, and 7. Prostaglandin F2α (PGF2α; Lutalyse™, Phamacia Co., Puurs, Belgium) was administered at 9 am and 9 pm, and the CIDR device was removed at 9 pm on day 6. The cows were administered 100 μg gonadotropin hormone-releasing hormone (GnRH; Fertagyl™, Intervet, Boxmeer, The Netherlands) at 9 am on day 8. Subsequently, semen from a Hanwoo bull was simultaneously used to artificially inseminate the cows at 9 pm on day 8 and 9 am and 9 pm on day 9 at 12 h intervals.

### 2.3. Blood Sampling and Analysis

Blood was collected from the jugular vein by venipuncture in the morning before feeding each day when the study began (day -7), when the CIDR was inserted (day 0), when the first AI was performed (day 8), and when the embryos were harvested (Day 16).

Blood profiles (total glucose, total cholesterol, high-density lipoprotein, low-density lipoprotein, total protein, and blood urea nitrogen (BUN)) were evaluated by a Cobas8000 C702 system with a blood analyzer (Roche Diagnostics, Basel, Switzerland) with the following assyas: TP Gen.2, UREAL, GLUC HK Gen.3. Electrochemiluminescence immunoassay (ECLIA, Roche) was used to detect estradiol and progesterone concentrations, which were analyzed using Cobas^®^ 6000 (Roche).

### 2.4. Statistical Methods 

Statistical analyses were performed using SPSS software (version 27.0; IBM Corp., Armonk, NY, USA). The Kruskal–Wallis test and the Mann–Whitney U with Bonferroni’s method were used. A generalized linear mixed model with Bonferroni correction was used to evaluate repeated measurements of blood profiles and estradiol and progesterone levels. In the generalized linear mixed model with Bonferroni correction, time and group were fixed effects, whereas cattle nested within the group were random effects. Data were expressed as mean ± standard deviation (SD). Statistical significance was set at *p* < 0.05 for the Kruskal–Wallis test and *p* < 0.017 (0.05/3) for the Mann–Whitney U with the Bonferroni method. Statistical significance for the Mann–Whitney U with the Bonferroni method was divided by three, as the groups could be compared in three different ways (CON-LC, CON-HC, and LC-HC).

## 3. Results

### 3.1. Descriptive Statistics and Results of Embryos According to Feeding Differences

When the embryos were harvested, the HC group had the highest BCS (3.25 ± 0.38), followed by the CON (3.22 ± 0.26) and LC (2.95 ± 0.16) groups (*p* = 0.043; Table 1), whereas body weight did not differ between the groups (*p* = 0.213). The difference in body weight from day -7 to day 16 was lowest in the LC group (−28.2 ± 15.9 kg), followed by the CON (−13.6 ± 16.4 kg) and HC (0.4 ± 27.3 kg) groups (*p* < 0.05). The number of embryos that were harvested was highest in the LC group (15.00 ± 6.82), followed by the HC (13.75 ± 7.85) and CON (6.22 ± 4.44) groups (*p* = 0.016). However, the number of transferable and transferable-to-harvesting embryos did not differ between the groups (*p* > 0.05).

### 3.2. Changes in Sexual Hormones and Serum Biochemical Parameters

In terms of estradiol and progesterone concentrations, the CON, LC, and HC groups did not differ from the start of the study to embryo harvesting (*p* > 0.05) (Figure 2). The estradiol concentration on day 8 in the LC group was the highest (160.9 ± 36.8 pg/mL), followed by the HC (129.0 ± 18.9 pg/mL) and CON (115.2 ± 36.8 pg/mL) groups; however, the result was not significant (*p* = 0.375).

The levels of glucose, total cholesterol, high-density lipoprotein, low-density lipoprotein, and total protein did not differ between the groups throughout the study (*p* > 0.05). The level of BUN differed on day 0 among the groups (*p* = 0.029); the HC group had the highest level of BUN (12.13 ± 1.81 mg/dL), followed by the CON (9.22 ± 3.07 mg/dL) and LC (9.10 ± 2.08 mg/dL) groups. The level of BUN did not differ among the groups until embryo harvesting.

## 4. Discussion

In this study, the effects of short-term concentrate intake on embryo harvesting, sexual hormones, and serum biochemical parameters were evaluated in indigenous Korean cows using estrus synchronization and superovulation. The LC group lost the most body weight and had the lowest BCS at the embryo harvest stage, followed by the CON and HC groups. While the groups in which the amount of formula feed was changed produced more embryos, the LC group had the highest number of harvested embryos, followed by the HC and CON groups. Sex hormone levels and serum biochemical parameters did not differ according to the amount of formula fed, except for a temporary increase in the HC group.

Nutritional provision influences the reproductive efficiency of cattle. Cattle that undergo long-term nutrient restriction enter anestrus, and the dominant follicle growth rate decreases [13]. Lowering feed intake for a short period before AI contributes to an improved conception rate. Previous studies have indicated that reducing feed intake after AI in cows, specifically in those that initially had a high feed intake, results in decreased embryo survival rates of approximately 40% [14]. Meanwhile, cattle that experience nutrient restriction in the short term produce more follicles and show improved embryo quality following superovulation [7]. However, feeding concentrate ad libitum has a detrimental effect on embryo yield and quality following superovulation in cattle [10]. In this study, regardless of a decrease or increase in the amount of concentrate, more embryos were harvested in the feed-intake-changing groups than the feed-intake-maintaining group. Similarly to the results of a previous study, nutrient restriction increased the number of embryos harvested [7]. Interestingly, a high concentrate feed intake also contributed to an increase in the number of harvested embryos. This might be due to differences in study design and the amount of concentrate provided to the cows. In this study, the cows were limited to a concentrate intake of 4 kg, whereas one study limited their cows to 3 kg and another provided the cows with concentrate ad libitum [10]. The previous study compared limited concentrate intake (3 kg) with concentrate fed ad libitum [10]. This indicated that an increase in concentrate could positively contribute to the number of harvested embryos, unless the increase was not severe. Furthermore, the number of transferable embryos was higher in the feed-intake-changing (LC and HC) groups, although the difference was not statistically significant. These findings suggest that changing the concentrate intake for a short time may contribute to more embryos being available for harvest. 

Circulating estradiol and progesterone concentrations are important for improving the reproductive performance of dairy and beef cattle. Knowledge of ovarian function is essential for understanding the secretion kinetics of the sex hormones, estradiol and progesterone [15,16]. Consistent with a previous study, in this study, the estrogen concentration reached a peak during the estrous cycle, which was similar to the onset of the luteinizing hormone (LH) surge [17]. This pattern of progesterone concentration is consistent with the results of previous studies [15], which have reported that the concentration of progesterone reaches its lowest level during AI and after FSH administration and then increases after ovulation. Circulating estradiol concentrations during pre-ovulation influence embryo survival [18]. Progesterone concentration is essential for oocyte and early embryo development [19]. In this study, the estradiol and progesterone levels were not significantly different among the groups at each stage of the experimental period. The estradiol and progesterone results might have influenced the transferable-to-harvesting embryo ratio, which did not significantly differ among the groups.

In cattle, some serum metabolites, including glucose, total protein, and BUN, are not associated with embryo production in superovulation [20,21]. In this study, the glucose and total protein concentrations did not differ at any stage. BUN concentrations were similar among the groups, although transitory differences were observed between the LC and HC groups when the CIDR was inserted; however, the difference was small. BUN levels are influenced by many factors, including liver synthesis and excretion via urea, saliva, and the gastrointestinal tract [22]. The transitory difference between LC and HC based on the amount of concentrate may be resolved by metabolic adjustments. The steroid hormones estrogen and progesterone, which are associated with fertility, are synthesized in ovarian cells using cholesterol [23]. The difference in the amount of concentrate in this study did not affect cholesterol concentration, which might have resulted in no difference in the estradiol and progesterone concentrations.

It was determined that changing the amount of concentrate over a short period of time contributed to embryo production in superovulated cows without changes in serum metabolites. An adequate increase in the concentrate amount had a positive effect on embryo production. However, this study has several limitations. First, we did not investigate the reason why a decrease and increase in concentrate intake for a short period led to a higher number of harvestable embryos. Furthermore, it was conducted with a small population size at one farm. The increasing feed intake groups did not vary; nevertheless, ad libitum feeding was reported to have a detrimental effect on embryo production during superovulation. In addition, potential impacts of changes in body weight and differences in BCS should have been investigated regarding the long-term reproduction and health implications for donor cows. Therefore, further studies are required in order to elucidate the mechanisms underlying the effects of short-term changes in concentrate intake on the number of harvestable embryos, embryo production, and long-term reproduction and health in superovulated cows.

## 5. Conclusions

The present study demonstrated that a short-period increase, as well as a short-term decrease in the amount of feed concentrate, could result in an increase in embryo production during superovulation in cows. Considering the lack of differences from the control group in circulating hormones and serum biochemical levels, changes in concentrate intake for a short time could be used to harvest more embryos without affecting the health of cows. These findings provide insights into the effects of nutrition on superovulation in cows. Further studies are required in order to elucidate the mechanism of embryo production by superovulation with increasing concentrate amounts.

## Figures and Tables

**Figure 1 animals-14-02591-f001:**
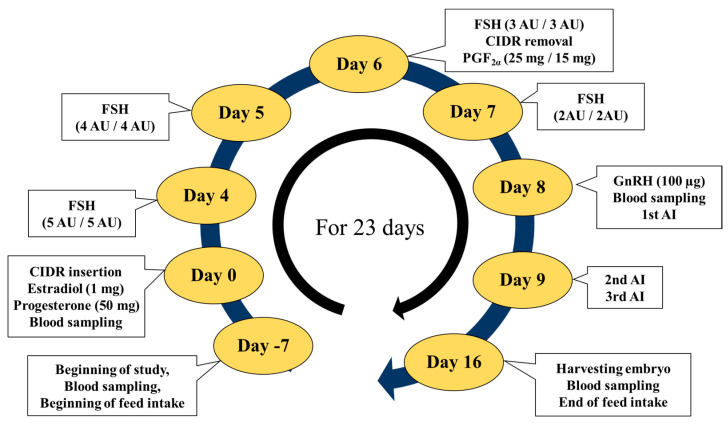
Study timeline indicating the nutritional and hormone treatments and blood sampling. CIDR, controlled internal drug release; FSH, follicular stimulation hormone; AU, armor unit; PGF2α, prostaglandin F2 alpha; GnRH, gonadotropin-releasing hormone; AI, artificial insemination.

**Figure 2 animals-14-02591-f002:**
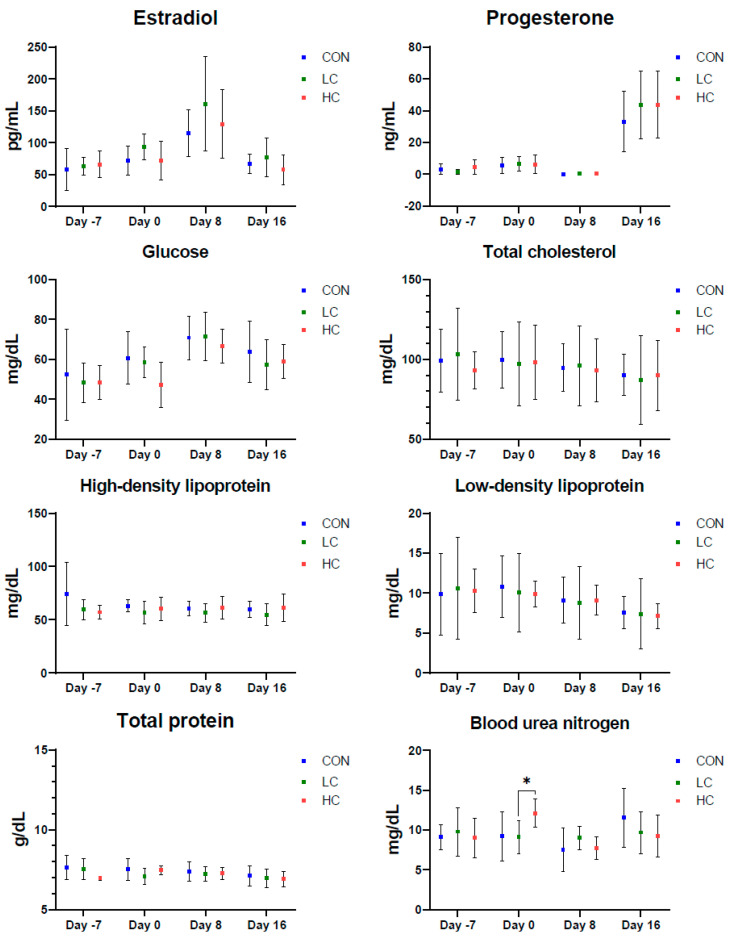
Effects of short-term nutritional differences on hormone and serum biochemical parameters. CON, control group; LC, low concentrate-fed group; HC, high concentrate-fed group. * *p* < 0.017 (Mann–Whitney U test with Bonferroni’s method for the Kruskal–Wallis test).

**Table 1 animals-14-02591-t001:** Descriptive statistics and results of harvesting and transferable embryos according to feeding differences.

Variable	CON	LC	HC	*p*-Value
Number	9	10	8	
Age, year	7.20 ± 0.82	7.27 ± 1.27	7.05 ± 1.13	0.684
Parity	2.11 ± 0.33	2.00 ± 0.47	1.88 ± 0.35	0.464
Days since calving	333.7 ± 94.3	273.8 ± 54.0	283.8 ± 50.6	0.371
Body Condition Score				
Beginning of study	3.28 ± 0.26	3.15 ± 0.24	3.38 ± 0.44	0.428
Harvesting embryo	3.22 ± 0.26	2.95 ± 0.16	3.25 ± 0.38	0.043
DBBH	−0.06 ± 0.17	−0.20 ± 0.26	−0.13 ± 0.23	0.370
Body weight, kg				
Beginning of experiment	452.9 ± 51.9	478.0 ± 46.5	502.0 ± 85.9	0.370
Harvesting embryo	439.3 ± 48.8	449.8 ± 47.1	502.4 ± 76.9	0.213
DBBH	−13.6 ± 16.4	−28.2 ± 15.9	0.4 ± 27.3	0.029
Number of harvesting embryos	6.22 ± 4.44 ^a^	15.00 ± 6.82 ^b^	13.75 ± 7.85 ^ab^	0.016
Number of transferable embryos	4.22 ± 2.82	9.90 ± 8.12	9.00 ± 6.55	0.169
Transferable-to-harvesting embryos	0.75 ± 0.23	0.65 ± 0.36	0.59 ± 0.24	0.583

CON, control group; LC, low concentrate group; HC, high concentrate group; DBBH, differences between at the beginning of study and embryo harvesting. Data are expressed as the mean ± standard deviation values. ^a,b^: Different letters in the same row indicate significant differences (*p* < 0.017, Mann–Whitney U test with Bonferroni’s method).

## Data Availability

The dataset generated and/or analyzed during the current study are available from the authors upon reasonable request.

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
