# Peer review of "Effects of Short-Term Differences in Concentrate Feeding on the Recovery of In Vivo Embryos in Hanwoo Donor Cows through Superovulation Treatment"

_animals, 2024, doi:10.3390/ani14172591_

Round 1

Reviewer 1 Report

Comments and Suggestions for Authors

Reviewer's comment to author

General Comments;

The study investigated the effects of short-term differences in concentrate feeding on in vivo embryo recovery in Hanwoo donor cows by superovulation treatment. The results showed that an important aspect of bovine reproduction was investigated and valuable insights were gained into the effects of variations in concentrate intake on embryo production. The manuscript is well organised and addresses an important gap in the literature. Although this study is interesting and the research design is considered to be good, the following points need to be corrected.

Major comments;

1)    The introduction to your manuscript seems a little weak. The introduction should help the reader to understand the need and value of the research by clearly explaining the background, significance, gaps in current knowledge and aims of the research. The introduction would be stronger if you added the following content.

a. Explain more specifically how feed nutrition affects the reproductive performance of livestock. For example, explain how energy deficiencies or excesses affect reproductive health, especially in certain breeds such as Hanwoo cows.

b. Clarify the differences between this study and previous studies; for example, it should be emphasised that previous studies have mainly focused on other breeds or long-term feeding changes, whereas this study investigates the effects of short-term concentrate feeding changes in Hanwoo.

c. Clearly state your research aims and hypothesis. For example, state a hypothesis such as "In the short term, increasing or decreasing concentrate intake will result in significant changes in embryo production".

2)    The sample sizes for the treatment groups (CON, LC, HC) are relatively small (n=9, n=10, n=8, respectively). Did these groups have a history of previous MOET attempts? If so, it would be nice to show comparative data.

3)    The manuscript mentions the small sample size as a limitation. It would be helpful to discuss other potential limitations, such as the duration of the study, environmental conditions, and the generalizability of the findings to other breeds or management systems.

4)    While the study mentions random division of cows into three groups, it is crucial to provide more details about the randomization process. How was randomization performed? Were there any measures taken to ensure that other factors (e.g., age, parity) were evenly distributed across groups? If possible, you should add something to your manuscript about this.

5)    The study concludes that both a decrease and an increase in concentrate intake led to a higher number of harvested embryos compared to the control. However, the mechanisms behind these findings are not fully explored. It should be discussed fully in the Discussion section.

Minor comments;

L14: eggs? Suggests changing to oocytes.

L45-46: We recommend that 'in vivo' and 'in vitro' be italicized.

L81-90: Absence of explanation of how to estrous synchronization. Please include a reference or explain in more detail. For example, in the case of FSH, it is not mentioned that it is injected twice a day, and in Figure 1. it is also difficult to understand because it is written as “2.5 mL/2.5 mL”. It is also written in mL instead of AU or mg, which is confusing for interpretation.

L91-94: Is the Hanwoo semen used in the AI the same semen and was it produced at the same time? If so, please explain this further in your manuscript.

L110: The use of the Mann-Whitney U test with Bonferroni's method for multiple comparisons is appropriate. However, the presentation of p-values and statistical significance could be clearer. For example, the threshold for significance after Bonferroni correction should be explicitly stated in the methods section.

L122-125: The study found that the LC group had the lowest BCS at embryo harvest. This finding suggests a potential negative impact of low concentrate intake on overall cow health, which is not thoroughly discussed. While the primary focus is on embryo yield, the long-term health implications for donor cows undergoing such dietary changes should also be considered.

L132-134: In Table 1, it is stated that different letters in the same row indicate significant differences, but some p-values are greater than 0.05 (e.g. parity, BCS at entry). This could be confusing and should be clarified.

L138: Was the estradiol / progesterone ratio not statistically significant?

Comments on the Quality of English Language

Needs some English proofreading, such as the singular and plural forms of expressions.

Author Response

General Comments;

The study investigated the effects of short-term differences in concentrate feeding on in vivo embryo recovery in Hanwoo donor cows by superovulation treatment. The results showed that an important aspect of bovine reproduction was investigated and valuable insights were gained into the effects of variations in concentrate intake on embryo production. The manuscript is well organised and addresses an important gap in the literature. Although this study is interesting and the research design is considered to be good, the following points need to be corrected.

Major comments;

1)    The introduction to your manuscript seems a little weak. The introduction should help the reader to understand the need and value of the research by clearly explaining the background, significance, gaps in current knowledge and aims of the research. The introduction would be stronger if you added the following content.

  1. Explain more specifically how feed nutrition affects the reproductive performance of livestock. For example, explain how energy deficiencies or excesses affect reproductive health, especially in certain breeds such as Hanwoo cows.

Response: Thank you for the specific suggestion. We have added information on the effects of nutrition on reproductive performance in lines 57 – 62.

  1. Clarify the differences between this study and previous studies; for example, it should be emphasised that previous studies have mainly focused on other breeds or long-term feeding changes, whereas this study investigates the effects of short-term concentrate feeding changes in Hanwoo.

Response: We have added some information clarifying the differences between our study and previous ones in lines 68–70.

  1. Clearly state your research aims and hypothesis. For example, state a hypothesis such as "In the short term, increasing or decreasing concentrate intake will result in significant changes in embryo production".

Response: Thank you for pointing this out. We have added our research aims and hypothesis in lines 71–75.

2)    The sample sizes for the treatment groups (CON, LC, HC) are relatively small (n=9, n=10, n=8, respectively). Did these groups have a history of previous MOET attempts? If so, it would be nice to show comparative data.

Response: The treatment groups did not have a history of previous MOET attempts. They were impregnated via artificial insemination.

3)    The manuscript mentions the small sample size as a limitation. It would be helpful to discuss other potential limitations, such as the duration of the study, environmental conditions, and the generalizability of the findings to other breeds or management systems.

Response: Thank you for the helpful suggestion. We have added other potential limitations.

4)    While the study mentions random division of cows into three groups, it is crucial to provide more details about the randomization process. How was randomization performed? Were there any measures taken to ensure that other factors (e.g., age, parity) were evenly distributed across groups? If possible, you should add something to your manuscript about this.

Response: The cows used in this study were raised in the same farm and fed the same formula feed and hay. Therefore, we considered them as similar and divided them into three groups according to their age and parity. Each group was randomly given different amounts of feed. We did not implement any other measures.

5)    The study concludes that both a decrease and an increase in concentrate intake led to a higher number of harvested embryos compared to the control. However, the mechanisms behind these findings are not fully explored. It should be discussed fully in the “Discussion section”.

Response: Thank you for the insightful comment. The concentration changes in estradiol, progesterone, and serum metabolites were investigated to clarify the mechanisms. However, their concentrations did not differ among the groups. Therefore, we have added the investigation of underlying mechanisms as a limitation of this study.

Minor comments;

L14: eggs? Suggests changing to oocytes.

Response: Thank you for the suggestion. We have changed eggs to oocytes.

L45-46: We recommend that 'in vivo' and 'in vitro' be italicized.

Response: We have italicized all Latin words in the text.

L81-90: Absence of explanation of how to estrous synchronization. Please include a reference or explain in more detail. For example, in the case of FSH, it is not mentioned that it is injected twice a day, and in Figure 1. it is also difficult to understand because it is written as “2.5 mL/2.5 mL”. It is also written in mL instead of AU or mg, which is confusing for interpretation.

Response: Thank you for the insightful comment. We used estrous synchronization protocols for Hanwoo cows used in previous studies (‘Historical perspectives and recent research on superovulation in cattle’ [Bo et al.] and ‘The outcome and economic viability of embryo production using IVF and SOV techniques in the Wagyu breed of cattle [Facioli et al.]). We have changed the units used in Figure 1 and the explanation in lines 89–104.

L91-94: Is the Hanwoo semen used in the AI the same semen and was it produced at the same time? If so, please explain this further in your manuscript.

Response: Thank you for the suggestion. We have changed “using semen from a Hanwoo bull” to “using semen produced from a Hanwoo bull simultaneously” in lines 102-103.

L110: The use of the Mann-Whitney U test with Bonferroni's method for multiple comparisons is appropriate. However, the presentation of p-values and statistical significance could be clearer. For example, the threshold for significance after Bonferroni correction should be explicitly stated in the methods section.

Response: Thank you for the suggestion. We have added some explanations in lines 128–131.

L122-125: The study found that the LC group had the lowest BCS at embryo harvest. This finding suggests a potential negative impact of low concentrate intake on overall cow health, which is not thoroughly discussed. While the primary focus is on embryo yield, the long-term health implications for donor cows undergoing such dietary changes should also be considered.

Response: Thank you for your insightful comment. We believed the reduction in body weight and the lowest BCS at embryo harvest in the LC group to be a natural process since there was no difference in serum metabolites. Therefore, we added the negative impact as a limitation in lines 233–235.

L132-134: In Table 1, it is stated that different letters in the same row indicate significant differences, but some p-values are greater than 0.05 (e.g. parity, BCS at entry). This could be confusing and should be clarified.

Response: Thank you for pointing this out. We wrote a-b: Different letters in the same row indicate significant differences (p < 0.017, according to the Mann-Whitney U test with Bonferroni’s correction) in lines 148–149. We only indicated a b in the row indicating the number of harvesting embryos, in which p-values were < 0.05. DBBH had p-value of 0.029, but the multiple comparison using the Mann–Whitney U test with Bonferroni's correction gave us a p-value > 0.017. Therefore, a b was not indicated.

L138: Was the estradiol / progesterone ratio not statistically significant?

Response: We analyzed estradiol/progesterone at each stage, but the ratios were not statistically significant. In addition, we analyzed estradiol on day 8 / progesterone on day 16 since they were high, but they were not statistically significant either.

Reviewer 2 Report

Comments and Suggestions for Authors

In the present work, Ha et al. try to study the effects of short-term differences in concentrate feed intake on in-vivo embryo production through superovulation in indigenous Korean cows. This study suggests that more embryos are harvested when short-term changes in concentrate intake are made during superovulatory responses in Hanwoo cows. However, there are some questions that should be explained.

Major concerns

1. This study showed that the number of harvested embryos was higher in the LC group (15.00±6.82) than HC (13.75±7.85) and CON (6.22±4.44) groups. In addition, CON group was fed 2.0 kg/day of formula feed formula, and LC group was fed 0 kg/day of formula feed formula, and HC group was fed 4.0 kg/day of formula feed. These results are complicated. Therefore, more information should be provided for cows fed. Is these cows were fed based on nutritional requirements standards? The nutritional composition for each group should be provided. In addition, the composition of formula feed was not included vitamins and minor elements.

2. More information about cows should be provided, including body weight, milk yield and milk production period, and so on. Some information about cows was present in Table 1 of result section, which should be moved to Materials and Methods section.

3. In Figure 2. At day -7, the levels of estradiol and progesterone were lower, and we want to know that these cows are at which stage of the estrous cycle?follicular phase or luteal phase?

4. English grammar and writing style should be checked and revised throughout the manuscript. I suggest a language revision before potential publication, or it is supported by a professional English language proofreading service.

Minor concerns

1. Simple summary should be rewritten, which is not a simple summary of this manuscript.

2. Introduction section should be rewritten, which should be focused on nutrition.

3. Materials and Methods section, more information about cows and feeding should be added.

4. Lines 201-203, ‘In cattle, some serum metabolites, including glucose, total protein, and BUN, are not associated with embryo production in superovulation. In this study, the glucose and total protein concentrations did not differ at any stage.’ Therefore, it is not necessary for detecting serum metabolites.

Comments on the Quality of English Language

English very difficult to understand/incomprehensible.

Author Response

In the present work, Ha et al. try to study the effects of short-term differences in concentrate feed intake on in-vivo embryo production through superovulation in indigenous Korean cows. This study suggests that more embryos are harvested when short-term changes in concentrate intake are made during superovulatory responses in Hanwoo cows. However, there are some questions that should be explained.

Major concerns

  1. This study showed that the number of harvested embryos was higher in the LC group (15.00±6.82) than HC (13.75±7.85) and CON (6.22±4.44) groups. In addition, CON group was fed 2.0 kg/day of formula feed formula, and LC group was fed 0 kg/day of formula feed formula, and HC group was fed 4.0 kg/day of formula feed. These results are complicated. Therefore, more information should be provided for cows fed. Is these cows were fed based on nutritional requirements standards? The nutritional composition for each group should be provided. In addition, the composition of formula feed was not included vitamins and minor elements.

Response: Thank you for the comment. As mentioned in lines 72–76, we randomly divide the cows into three groups according to feed formula intake. The CON group was fed 2.0 kg/day of feed formula, and LC group was fed 0 kg/day of feed formula, and HC group was fed 4.0 kg/day of feed formula. We performed the same estrous synchronization and artificial insemination procedures on all cows the three groups and found that the number of harvested embryos was higher in the LC group (15.00±6.82) than HC (13.75±7.85) and CON (6.22±4.44) groups. We provided the nutritional information of the feed in lines 76–78. However, we could not include vitamins and other minor elements, although in hindsight this should have been included. We cannot present more information about the feed formula than that mentioned in lines 76–78.

  1. More information about cows should be provided, including body weight, milk yield and milk production period, and so on. Some information about cows was present in Table 1 of result section, which should be moved to Materials and Methods section.

Response: Thank you for the comment. Beef cows were used in this study; therefore, milk yield and milk production period were not required. This manuscript is for communication, so that we should have at least figures and tables. If we include another table for age and parity, it would be meaningless.

  1. In Figure 2. At day -7, the levels of estradiol and progesterone were lower, and we want to know that these cows are at which stage of the estrous cycle?follicular phase or luteal phase?

Response: Thank you for the suggestion. We did not investigate the stage of the estrous cycle on day 7. However, considering the concentration of estradiol and progesterone, the cows were in the follicular phase. Estradiol was high on day 8, which is when we performed the first AI. Progesterone was high on day 16 which is when ovulation was complete.

  1. English grammar and writing style should be checked and revised throughout the manuscript. I suggest a language revision before potential publication, or it is supported by a professional English language proofreading service.

Response: Thank you for your concern. We have sent the manuscript for English editing before submission, and it will be revised again after all reviewer suggestions are implemented.

Minor concerns

  1. Simple summary should be rewritten, which is not a simple summary of this manuscript.

Response: Thank you for the suggestion. We have rewritten the simple summary to be more descriptive of our study (lines 14–21).

  1. Introduction section should be rewritten, which should be focused on nutrition.

Response: Thank you for the suggestion. We have added information on the effects of nutrition on reproductive performance in lines 57 – 62.

  1. Materials and Methods section, more information about cows and feeding should be added.

Response: Thank you for the suggestion. We would like to clarify that the cows used in our study were beef cows, not dairy cows. Additionally, the information regarding the feeds that were offered to the cows is documented in the manuscript.

  1. Lines 201-203, ‘In cattle, some serum metabolites, including glucose, total protein, and BUN, are not associated with embryo production in superovulation. In this study, the glucose and total protein concentrations did not differ at any stage.’ Therefore, it is not necessary for detecting serum metabolites.

Response: Thank you for your insight. We wanted to clarify that short-term changes in concentrate intake did not affect the nutritional metabolism of cows. Fellow scholars might be curious if the results of serum metabolites were not included. Therefore, we believe it would be better to show the results of serum metabolites.

Reviewer 3 Report

Comments and Suggestions for Authors

Sent in email

Comments on the Quality of English Language

Sent in email

Author Response

Thank you for the comment. We have changed our manuscript.

Round 2

Reviewer 1 Report

Comments and Suggestions for Authors

I appreciate the thorough revisions you have made to your manuscript in response to my comments. Your efforts have significantly improved the quality and clarity of your study on the effects of short-term differences in concentrate feeding on in vivo embryo recovery in Hanwoo donor cows. The manuscript is well organised and addresses an important gap in the literature on bovine reproduction and nutrition. The revisions have improved the clarity and depth of the manuscript, making it a valuable contribution to the field.

Author Response

Thank you for your thorough review and constructive feedback on our manuscript. 

Reviewer 2 Report

Comments and Suggestions for Authors

Thanks for author’s responses. However, some questions should be paid attention.

1. Short-term differences in concentrate feed intake has directly effects on the results of this study, but the authors deny to provide the details of vitamins and other minor elements. Vitamins and other minor elements have essential effects on animal growth and development. Is these cows were fed based on nutritional requirements standards?

2. As a Communication, authors reported that some serum metabolites, including glucose, total protein, and BUN, are not associated with embryo production in superovulation. Therefore, these serum metabolites may be deleted.

3. Information about cows, including body weight, milk yield and milk production period, should be moved to Materials and Methods section. If this information has significant difference, this study will be meaningless. This information belongs to the experimental design.

4. In this study, there are only numerical results present. We do not think that this manuscript is suitable for publication in this Journal (Animals).

Comments on the Quality of English Language

Extensive editing of English language required.

Author Response

Response to Reviewer 2 Comments

  1. Short-term differences in concentrate feed intake has directly effects on the results of this study, but the authors deny to provide the details of vitamins and other minor elements. Vitamins and other minor elements have essential effects on animal growth and development. Is these cows were fed based on nutritional requirements standards?

Response: Thank you for your thorough review and constructive feedback on our manuscript. We appreciate your observation regarding the importance of vitamins and other minor elements in animal growth and development, and we acknowledge that a more detailed analysis of these components would have enriched our study. While our primary focus was on the short-term effects of concentrate feed intake on specific serum metabolites, we agree that a comprehensive analysis including vitamins and minor elements could have provided a deeper understanding of the nutritional influences on the results. Unfortunately, owing to limitations in our study design and the available resources, we were unable to include detailed analyses of these components. We agree that future studies should aim to incorporate a more thorough examination of feed composition, including vitamins and minor elements, to better understand their effects and ensure that the nutritional requirements of the animals are fully met. Such an approach would undoubtedly offer valuable insights into the complex interactions between nutrition and reproductive performance. Once again, we appreciate your feedback and will consider including this important aspect in our future research endeavors.

  1. As a Communication, authors reported that some serum metabolites, including glucose, total

protein, and BUN, are not associated with embryo production in superovulation. Therefore, these serum metabolites may be deleted.

Response: Thank you for your insightful review of our manuscript. As you pointed out, our findings indicate that certain serum metabolites, including glucose, total protein, and blood urea nitrogen (BUN) do not show a significant association with embryo production in superovulation. We agree that these metabolites may be excluded from the direct analysis of embryo production. However, the primary aim of our study extends beyond simply identifying direct associations between serum metabolites and embryo production. We seek to explore how changes in the intake of concentrated feed affect physiological changes in the body, and how these changes, in turn, might influence embryo production. Variations in concentrated feed intake can have a significant impact on metabolic processes within the body, potentially altering levels of glucose, total protein, and BUN. Understanding how these physiological changes affect embryo production is central to our research. Therefore, while glucose, total protein, and BUN may not be directly associated with embryo production, they serve as important indicators of physiological changes resulting from variations in concentrated feed intake. From this perspective, our study aims to clarify the interaction among feed management, metabolic changes, and embryo production, ultimately providing practical strategies for improving reproductive performance. Based on your comments, we will work to further clarify the focus of our study and ensure a thorough analysis of the results, enhancing the scientific contribution of our research.

  1. Information about cows, including body weight, milk yield and milk production period, should be moved to Materials and Methods section. If this information has significant difference, this study will be meaningless. This information belongs to the experimental design.

Response: Thank you for your insightful comments on our manuscript. Following your suggestion, we have moved the information regarding body weight and other relevant details to the Materials and Methods section in lines 95–104, as it indeed pertains to the experimental design. However, we would like to clarify that this study was conducted using Hanwoo cattle, which are beef cattle, not dairy cows. Therefore, there is no information on milk yield or milk production period in our study, as these variables are not applicable to this particular research. We have instead included the "days since calving" indicator in Table 1. We appreciate your attention to detail and hope this clarification helps to better understand the scope and context of our work.

  1. In this study, there are only numerical results present. We do not think that this manuscript is suitable for publication in this Journal (Animals)

Response: Thank you for your review of our manuscript and for providing your candid feedback. We understand your concern regarding the presentation of the results in our study. While the manuscript does focus on the quantitative analysis of the data, we believe that these numerical results are crucial for understanding the impacts of the variables studied. The data presented offer valuable insights into the specific research questions we aimed to address. We have endeavored to present the findings in a clear and objective manner, with the goal of contributing to the broader scientific understanding within the field of animal science. We respectfully suggest that the numerical results, although perhaps less narrative in nature, still provide significant scientific value, particularly for studies that are driven by precise, data-driven conclusions. We are open to further enhancing the manuscript by expanding the discussion section to provide more context and interpretation of these results, if that would align better with the expectations of the journal. We appreciate the journal’s high standards and hope that, with these considerations, the manuscript might still be considered for publication. We are willing to make any additional revisions that could improve the  suitability of the manuscript for Animals.